# Reviewing Simulation Technology: Implications for Workplace Training

**Diana R. Sanchez** [1,*]**, Amanda Rueda** [1]**, Kentaro Kawasaki** [1]**, Saar Van Lysebetten** [2] **and Daniel Diaz** [1]

1   Psychology Department, Ethnic Studies & Psychology Building, San Francisco State University,
    1600 Holloway Ave, San Francisco, CA 94132, USA
2   Campus UZ Gent, Corneel Heymanslaan 10-Ingang 17, 9000 Gent, Belgium
*   Correspondence: sanchezdianar@sfsu.edu

**Abstract:** Organizations have maintained a commitment to using simulation technology for training purposes because it prepares employees for realistic work scenarios they may encounter and provides a relevant method for teaching hands-on skills. One challenge that simulation technology has faced is the persistent threat of obsolescence, where investment in an up-to-date solution can rapidly become irrelevant in a matter of months or years as technology progresses. This can be particularly challenging for organizations who seek out the best solutions to help develop and train employees while facing the constraints of limited resources and lengthy acquisition times for tools and equipment. Some industries and organizations may benefit from anticipating which technologies might best serve employees and stakeholders in the future. In this manuscript, we took a historical approach, looking at the history of training and the use of simulation-like experiences over time, which helps us identify historical themes in workplace training. Next, we carried out a systematic review of the recent training research using simulation technology to understand how these recent findings help us understand the identified historical themes. Lastly, we summarized the research literature on simulation technology used for training, and highlighted future directions and made recommendations for practitioners and researchers.

**Keywords:** simulation technology; workplace training; workplace preparation





## 1. Introduction

Simulation technology lends itself to training and development in the workplace, given the replication of real-world scenarios [1], which allows users to experience real-life situations and grow valuable skill sets while controlling the characteristics and environment of the learning experience [2]. This provides a uniquely valuable tool for organizations, because it offers opportunities not available in other training modalities (e.g., lectures, role plays, computer-based training [3]). For example, early simulation technology for training airplane pilots used ropes and pulleys to replicate turbulence during flight [4]. Simulation technology is evident in virtualized experiences as a way to enhance the realism of the experience [5], in addition to collecting objective metrics, providing immediate feedback, and capturing performance and assessment information [6]. Additionally, simulation technology may help organizations to better prepare employees to enter the workplace [7] due to the increased capability for providing work-related content, hands-on skills, and experiential learning [8,9]. Organizations have prioritized the transfer of usable skills from a training setting into a workforce setting, as seen in 2021 when large organizations reported spending over USD 17.5 million on workplace training despite the resource limitations of the pandemic [10]. Workplace preparation is critical for employees, particularly those who are early in their career. This importance is reflected in large organizational investments in training [11,12]. However, simulation technology is under an ever-present threat of obsolescence [13,14], and there can be concern for the future relevance of an investment when

considering the adoption of new or upgraded simulation technology for training [7,15]. This creates an understandable need to anticipate the future value of simulation technology for training. In the current manuscript, we attempt to identify a framework for understanding simulation technology by first reviewing the historical progression of workplace training as it has contributed to the development of simulation technology. In this review, we identified historical themes that emerged over time to provide a clearer understanding of the current research literature. Next, we performed a systematic review of recent simulation technology research that has been used for workplace training. From this, we provided a summary of the current research literature to demonstrate the relevance and distinctions of simulation technology and deepen our discussion of training applications. We next discussed the future directions of simulation technology, by discussing the current research streams that may continue to develop over the next several years and may contribute to predicting how simulation technology may advance.

### 1.1. Historical Themes of Workplace Training

Here, we discuss six historical themes of workplace training which contextualize the evolution of simulation technology. We present salient characteristics with respect to workplace training, and provide examples of milestones (i.e., social, scientific, and technological events or innovations) within each theme. The historical themes are generally chronological, but characteristics span time and geography. As such, the discussions and milestones may overlap time periods; see Table 1. We conclude each theme with a discussion of the influence on simulation technology. This historical review provides an interdisciplinary understanding of simulation technology used in workplace training. For a detailed review of workplace training specifically, see [16].

**Table 1.** Summary of Historical Themes in Workplace Training.

| | Social/Contextual | Scientific | Technological |
|---|---|---|---|
| Value of Knowledge | • Birth of philosophy<br>• Establishment of first university | • Legal restrictions on research<br>• Operationalizing lifelong learning | • Skilled workforce becomes a competitive advantage |
| Democratization of Knowledge | • Catalyzed desire for knowledge<br>• Just-in-time learning | • Renaissance era research | • Invention of printing press<br>• Virtualized information sharing |
| Science of Learning | • WWI/II changed workforce needs | • Standardized behavioral research<br>• Development of learning principles | • Industrial Revolution standardized work tasks |
| Scaling Productivity | • Establishment of labor skill levels | • Formalized workplace research | • Yellow Cab Company used one of first simulations |
| Emergence of Knowledge Work | • Growth of knowledge work<br>• Global/dynamic workforce | • Job characteristics theory<br>• Emphasis on culture/wellbeing | • Reliance on information and communication technology |
| Individualized Learning | • Adoption of peer-to-peer learning<br>• On-the-job training | • Shift towards 3rd generation instructional model | • Use of AI, VR, and computational psychometrics in instructional systems design |

### 1.1.1. Value of Knowledge

The first historical theme represents humans' innate desire for sense-making [17]. The birth of philosophy triggered tensions among intellectual and spiritual leaders, which

ultimately ignited academic inquiry. Scholars of science, literature, and philosophy in ancient Greece (600–300 BC) were treated as a threat to religious doctrine [18–20]. At a societal level, there were political movements to obstruct scientific study and access to information. This included legislation which restricted the study of anatomy using human cadavers in ancient Rome (753 BC–476 AD; [21,22]). Escalations over the value of intellectual knowledge catalyzed academics' resolve to seek knowledge. For example, Claudius Galenus (129–217 AD) examined animal cadavers to circumvent legislative restrictions on anatomical science [21]. The historical theme therefore exemplifies the consequences of intellectual curiosity, and the high stakes of learning.

The value and stakes of learning may be reflected in the competitive advantage of employing a knowledgeable and skilled workforce. Workplace training is considered a benefit to individual employees which could contribute to their advancement or promotion [23]. Training researchers have advocated for the integration of lifelong learning into talent development, which creates informal, intentional, and self-directed opportunities for learning [24]. Employees also demonstrate greater interest in companies with strong learning economies that promote their development over time [25], leading organizations to seek innovative ways to keep employees engaged in learning. Simulation technology is one method consistently utilized by organizations to expand their learning opportunities for employees [26,27].

### 1.1.2. Democratization of Knowledge

The increased value of knowledge led to an inevitable era of increasing the dissemination of knowledge. Technological developments such as the printing press during the Renaissance (1300–1700) increased public accessibility to information. Spiritual advocates sought to re-establish influence by controlling information, such as book bans from the Roman Catholic Church when content was perceived as heretical or lascivious [28]. Attempts at censorship engendered the democratization of knowledge. The Renaissance promoted the open pursuit of intellectual information (e.g., Leonardo da Vinci, 1452–1519, Nicolaus Copernicus 1473–1543, Galileo Galilei, 1564–1642), and the mass distribution of information contributed to a social shift from knowledge being exclusive to scholars, to being shared with the public.

The continued pursuit of democratized knowledge has influenced innovative systems for sharing information. Digitization has simplified the means of documenting, maintaining, and disseminating knowledge in organizations. For example, learning management systems and other digital platforms can lower the cost of traditional training programs (i.e., synchronous, in-person; [29]). Modern calls for equity have shaped continued efforts to make knowledge accessible, with many organizations placing workplace training curricula in open access repositories for employees [30]. By making content readily accessible, organizations may optimize when and how knowledge is provided, providing learners with relevant support when they need it [31]. Simulation technology can also provide just-in-time training solutions with immersive, hands-on experiences available to practice virtually for learners when needed. In the healthcare industry, simulations have been used to develop skills such as clinical decision-making, assessing risks, and refresher training on surgical procedures. Further, simulations used in conjunction with pre-briefing and repeated scenarios have been linked to improved learning outcomes for nurses [32].

### 1.1.3. Science of Learning

Interest in knowledge acquisition is evidenced by prolific research in human memory and pedagogy [33–36]. Formalized research on the mechanisms of learning have brought learning to the forefront of scientific interest. With the standardization of scientific methods [37], researchers examined new facets of learning including the mental processes associated with memory [38,39] and concepts such as observational learning and classical conditioning [40,41], which inform underlying principles used in workplace training today.

The fundamental concepts developed during the time of formalized research in learning and pedagogy have informed the primary frameworks, theories, and principles applied in formal workplace training today. For example, learner control is a broad term for various instructional design techniques and may be effective for learning through repeated practice [42]. Learner control is one stream of research which has influenced an early adoption of computer-based and simulation technology used for training [43]. Technology-based training is ubiquitous in today's workplace training environments, and good learning programs are based on robust, scientific learning principles (e.g., immediate feedback, practice opportunities) [14].

### 1.1.4. Scaling Productivity

A salient theme in modern history has been a focus on maximizing productivity through the application of learned knowledge, skills, and behaviors relevant to the job [34]. This has been important for organizations aiming to improve work-related skills through formalized training programs. One of the earliest pioneers of labor skills was Adam Smith [44], who introduced classifications of labor (common, skilled, qualified) which were characterized by the complexity of skills involved with the tasks. A need for efficient workers (i.e., WWI, the Industrial Revolution) [45] and burgeoning scientific interest in studying the world of work (Frederick Winslow Taylor, 1856–1915, Munsterberg, 1863–1916, Lillian Gilbreth, 1878–1972) created a fertile environment to study large-scale productivity in the workplace [18,46,47]. Although there was some decline in progress during the Great Depression [48], as jobs became increasingly complex and dangerous, organizations found new ways to assess and train workers. For example, in 1925 the Yellow Cab Company in Pittsburg commissioned one of the first uses of simulation technology for training to assess applicants on switchboard reaction time [49]. Interest in simulation technology for training was renewed during WWII due to the high-stakes nature of military training [50,51].

With the emphasis on productivity, a realization emerged that traditional training methods were not always sufficient for intricate task work, and simulation technology presented a viable alternative because it can create levels of complexity which mirror the real world. For example, officers can be placed in a hostage situation where they need to both verbally negotiate and physically react to a simulation of a perpetrator projected on a screen [52]. Firefighters may need to practice navigating a burning building, surveying safety risks, and completing tasks with limited visibility from equipment and smoke [53]. Simulating these complex conditions in replicated environments that can be customized to the unique needs of the learner allows intricate learning to take place within the context of safe, controlled environments [54].

### 1.1.5. Emergence of Knowledge Work

In addition to the growing complexity of work-related tasks, the emergence of knowledge work has placed technology at the center of the workforce and organizations expect employees to have a basic level of skill in using information and communication technologies (ICTs) [55]. Knowledge work encapsulates types of jobs in which cognitive processes (rather than physical tasks), autonomy, and ICTs are fundamental to performance. Knowledge work has shifted perspectives on technology, moving from technology as a perceived means for optimizing work to regarding technology as a fundamental tool for completing work. The ever-increasing globalization of work has ushered in a new conceptualization of where, when, and how individuals work. Flexible work locations and instant access to information is increasingly present in dynamic work environments [55,56]. Frequent changes in organizational structure (e.g., mergers, downsizing) and unpredictable environments require adaptability from both employees (e.g., being adaptable and resilient) [11] and employers (e.g., evolving workplace practices and services to employees). Employees expect more from workplace culture in terms of the experience they have at work and the degree of support they receive for their wellbeing. Reference [57] suggested that job-related factors influence an individual's psychological states (e.g., well-being), which in turn plays

a role in their job performance, degree of satisfaction, and work motivation. Workplace training is one way that organizations can develop the knowledge and skills which may help employees feel an increased sense of autonomy, reduce ambiguity and conflict in their role, and improve manager feedback practices.

Simulation technology has been advanced by two factors related to the emergence of knowledge work: the increasing availability of advanced tools, and the globalization of the modern workforce. The growing development and accessibility of tools (i.e., asynchronous communication and collaboration platforms, company intranets) has made simulation technology easier and more affordable for institutions to implement [58]. This offers a potential efficiency gain for organizations, particularly when compared to the financial costs and time costs associated with bringing together geographically dispersed teams for traditional, in-person training solutions. Ultimately, advancements in simulation technology have been accelerated by innovative technologies that allow organizations to remain competitive.

### 1.1.6. Individualized Learning

The shifting nature of work has resulted in employees expecting individualized care, attention, and resources provided by their organization [59]. For example, there is an increased desire for and availability of one-on-one experiences such as mentoring and coaching resources for companies [60]. Mentoring offers an individualized learning experience in which experienced professionals offer guidance and support to novice employees [61]. Executive coaching is a type of leadership development that is characterized by a high contextual-sensitivity coaching approach due to the complex roles and relationships among executives' stakeholders [62]. Despite the benefits of these one-on-one methods, they can come at a high cost [63]. An alternative method that organizations and institutions have tried implementing is alternative peer-to-peer learning experiences, which may take the form of on-the-job training wherein a novice and an expert are paired together [64]. Although the individual in the expert role may have limited motivation, skills, or knowledge, learners tend to enjoy the individualized attention in these interactions [65]. Although these are not new learning modalities, they have been facilitated by the increasing availability of virtualized learning platforms [66]. This shift from instructor-based learning to interaction-based learning (i.e., third-generation instruction) has challenged researchers and practitioners to rethink how training methods support learning [66]. Interaction-based learning takes a social constructivist view and highlights the importance of the learning environment and social context where learners drive their experience, thus facilitating individualized learning to a higher degree because of the unique set of knowledge, skills, and abilities that each learner chooses to engage in.

With a shift towards individualized experiences, simulation technology can be highlighted as it can provide individualized learning experiences at scale across large groups of geographically dispersed individuals [1]. The automation and customization of simulation technology can include collecting objective metrics on learners and feedback mechanisms (i.e., scoring features) customized to a user, or detailed and customized opportunities to practice. These features exceed what may be possible from a human instructor. The primary drawback to simulation technology for workplace training is the upfront cost and expertise required to design the programming needed to leverage these advanced technologies appropriately (e.g., [67]).

A summary of historical themes is provided in Table 1.

## 2. Method

*Current Research on Training Simulation Technology*

In this section, we provide a systematic review of the current research literature on simulation technology that has been used for workplace training in order to prepare people for the skills and tasks required on-the-job. The following criteria were used to select papers:

- Peer-reviewed journal articles published since the year 2000.

- Samples of working-age individuals between ages 18 and 65.
- Methodology had to include simulation technology which was intended to prepare individuals for work. This means that the simulation technology needed to replicate a real-world work environment for the purpose of training an individual.
- Had to include a virtualized component.

The methodology inclusion criteria were particularly important as the current paper aims to provide an overview of current simulation technology used for workplace training. Research papers which used participants that would be subject to specific employment laws and regulations (e.g., protected classes, neurodiverse populations, etc.) were excluded as this study aims to provide a review of simulation technology in standardized workplace training contexts.

Papers were identified through searches of the following electronic databases: Academic Search Ultimate, APA PsychInfo, Applied Science & Technology Source, Academic Search Premier, and Business Source Premier. The most recent search was performed in April 2023 using the following boolean search terms: Title/Abstract contains ("simulation technology" OR "simulation training" OR "work simulation" OR "workplace simulation" OR "job simulation") AND ("workplace training" OR "workplace learning" OR "job training" OR "job learning" OR "organization* training" OR "organization* learning").

### 3. Results

#### 3.1. Summary

The result initially provided us with 64 articles. We filtered those according to the criteria above and found 18 papers that met our requirements. This final selection of papers covered the use of simulation technology intended to develop work-relevant skills and abilities. The literature review is categorized in Table 2 according to the simulation type, subject matter, and outcome measures.

**Table 2.** Overview of existing simulation technology literature.

| Article | Year | Simulation Type | Subject Matter | Outcome Measures | Sample | Participants |
|---------|------|-----------------|----------------|------------------|--------|--------------|
| [68] | 2005 | Computer-driven simulation | Medical | Job-related task performance specific to interrupted suture score | 11 | Postgraduate Medical Residents |
| [69] | 2005 | Virtual Reality-based dental training | Dental | Job-related task performance | 42 | Dental Students |
| [70] | 2006 | Virtual Reality training simulator | Medical | Job-related task performance specific to carotid angiography | 20 | Interventional Cardiologists |
| [71] | 2007 | Aerospace simulator | Aerospace | Job-related task performance, Communication, Teamwork and Reactions | 29 | Space Shuttle MMT Members |
| [72] | 2009 | Web-based work simulation | Business | Performance as moderated by self-reflection | 360 | Employees |
| [73] | 2012 | Computer-based training simulator for industrial machinery | Construction | Reactions | 56 | Participants |
| [74] | 2012 | Virtual Reality simulation with haptic properties | Medical | Job-relevant skills and reactions | 10 | Undergraduate Students |

**Table 2.** *Cont.*

| Article | Year | Simulation Type | Subject Matter | Outcome Measures | Sample | Participants |
|---|---|---|---|---|---|---|
| [75] | 2013 | Business-simulation computer game | Business and Leadership | Reactions to leadership behaviors | 26 | Graduate Students |
| [76] | 2015 | Non-interactive 10 min 3D video | Medical | Empathy (modified Kiersma–Chen empathy scale) | 460 | Undergraduate Nursing Students |
| [77] | 2015 | High-technology human patient simulator (HPS) | Medical | Patient assessment skills | 101 | Undergraduate Students |
| [78] | 2016 | Safety and security training simulator for ship handling | Public Safety | Job-related performance | 14 | Students |
| [79] | 2016 | Flight simulator | Aerospace | Job-related task performance in flight abilities | 29 | Students |
| [80] | 2017 | Medical manikin | Medical | Reactions | 17 | Doctoral Students |
| [81] | 2019 | Healthcare interactive virtual simulation training system (HH-VSTS) | Medical | Job-related task performance specific to hazard management | 74 | Healthcare Workers and Students |
| [82] | 2019 | Maritime education and training (MET) Simulator | Maritime | Confidence in task performance | 11 | Students |
| [83] | 2019 | Medical manikins | Medical | Comfort with medical interventions | 57 | General Medicine Officers |
| [84] | 2020 | Medical manikin | Medical | Interteam communication | ~26 | Four Neurosurgeon teams; four medical student teams |
| [85] | 2020 | Experiential gamified simulation (WAGES-Business) | Business | Acknowledgement of unconscious bias | 126 | Undergraduate Students |

As is evidenced by the findings of our systematic literature review, simulations are used primarily for complex skills and tasks. Below is a general summary of the articles we found. The articles spanned publication from 2005 to 2020. Ten of the articles focused on dental or medical procedures [68–70,74,76,77,80,81,83,84]. Thirteen of the articles used students or recent graduates at least in part for their sample [68,69,74–82,84,85]. Ten of the studies directly measured job-related skills or tasks in some way as an outcome measure [68–72,74,77–79,81]. Five measured reactions to the simulation as part of their outcomes.

Based on our review of this literature, we provide first a summary distinguishing different types of training technology, followed by an overview of the identified benefits and challenges of using simulation technology.

A Summary of Current Simulation Technology

What we found in our reading of the current research was that as the workplace training landscape continues to evolve, simulation technology has transitioned from primarily manual mechanisms [4] to synthetic learning environments [3]. Within synthetic learning environments, simulation technology has emerged as a training medium that creates, extends, and manages learning objectives [86]. Simulation technology advances classic observational learning in simulations by aiding realism and embedded instruction [40,86]. Thus, simulation technology enables learners to interact with multi-faceted, complex issues where they can apply prior knowledge and skills to real-world issues related to their discipline [87].

### 3.2. Similarities and Distinctions from Related Training Approaches

Overlapping conceptualizations of training methods such as games or roleplays are common in workplace training research [6,88]. The application of simulation technology in workplace training can be distinguished by the degree of realism (i.e., fidelity) embedded into the design and delivery [81,89]. Additionally, simulation technology used in workplace training consistently takes place in scenario-based environments, where individuals interact with the environment to apply prior knowledge and practical skills [87,90,91]. Although simulation technology used for workplace training is distinct from other training methods, elements of synthetic and scenario-based learning environments can be used with many delivery methods. We offer a brief overview of training methods commonly addressed with simulation technology for workplace training in overlapping terms: serious games, game-based learning, and role plays. Table 3 highlights similarities and distinctions between these terms. Table 3 is not an exhaustive overview, but is intended to illustrate where overlap occurs in these training modalities.

**Table 3.** Overview of Terminology Related to Training Simulations.

| | Learning Objectives | Instructional Principles | Design Attributes | Commonly Used Context | Delivery Media |
|---|---|---|---|---|---|
| Simulation | • Transfer<br>• Self-Efficacy<br>• Teamwork Skills<br>• Procedural Knowledge<br>• Motivation<br>• Communication Skills<br>• Perceptual Knowledge<br>• Psychomotor Skills | • Scaffolding [2]<br>• Repetition [4]<br>• Motivating Learners [3]<br>• Variability of Roles, Responsibilities, Strategies, etc. [1]<br>• Error Management [1]<br>• Adaptive Difficulty [2] | • Immersion [t]<br>• Rules/Goals [t]<br>• High Fidelity<br>• Scenario-Based<br>• Gamification *<br>• Gameful Design * | • Healthcare<br>• Engineering<br>• Aerospace<br>• Law Enforcement<br>• Physics | • Face-to-Face<br>• Computer-based<br>• Web-based<br>• Virtual Reality<br>• Augmented Reality<br>• Mixed Reality<br>• Wearable Tech |
| Serious Games | • Transfer<br>• Self-Efficacy<br>• Teamwork Skills<br>• Procedural Knowledge<br>• Motivation<br>• Declarative Knowledge | • Scaffolding [2]<br>• Motivating Learners [3]<br>• Metacognitive Prompts [3]<br>• Contiguity [5]<br>• Pretraining [5] | • Action Language [t]<br>• Assessment [t]<br>• Conflict/Challenge [t]<br>• Control [t]<br>• Environment [t]<br>• Game Fiction [t]<br>• Human Interaction [t]<br>• Immersion [t]<br>• Rules/Goals [t]<br>• Gamification *<br>• Gameful Design * | • Education<br>• Business<br>• Mathematics<br>• Science<br>• Technology<br>• Healthcare<br>• Business | |
| Role Play | • Transfer<br>• Self-Efficacy<br>• Communication Skills | • Scaffolding [2]<br>• Repetition [4] | • Scenario-Based<br>• Gamification *<br>• Gameful Design * | • Business<br>• Language<br>• Leadership | |

Note: Bolded values are more salient to the given methodology. [t] [26] [1]—Effective Practice; [2]—Optimize Sequencing; [3]—Engage learners in their own learning; [4]—Develop past initial mastery; [5]—Organize content (Adapted from previous research [16]). * Gamification and gameful design employ one or more game design attributes to develop game-based learning interventions, whereas serious games consistently use all game design attributes, albeit in different ways or to varying degrees.

### 3.2.1. Serious Games

Serious games are games used specifically for educational rather than entertainment purposes [88]. Conversely, simulation technology may be used for non-educational purposes such as pure games (e.g., [92]) and economic simulations [93]. Beyond purpose, simulation technology and serious games can use game attributes differently in their design [26,27]. Serious games employ a complete set of attributes associated with game development, whereas simulation technology for training may use a subset of game attributes without taking the form of a game [75,88]. For example, both serious games and simulation technology for training include clear rules and goals of the mission to be achieved [26,94]. Similarly, the game attribute of immersion enables complex technical and social interactions that would occur in real-life to take place within both games and simulation technology used for training [95]. For example, a military game would likely use the context of basic exercise drills or missions [83] rather than an office setting to create an immersive environment.

### 3.2.2. Game-Based Learning

Game-based learning is the use of game-based technology to deliver, support, and enhance teaching, learning, assessment, and evaluation [96]. Unlike serious games (in which the game delivers the learning content), in game-based learning the instructor remains the primary provider of learning content and uses game elements to increase learner motivation [88]. To this end, instructional designers may employ different strategies to embed game-like elements into learning content or delivery methods, including simulation technology. One strategy, gamification, involves incorporating specific game attributes into existing training, typically to improve learner engagement [95]. Similarly, gameful design places game attributes at the center of the development process for new training or assessment instruments [97]. A game can be set within the context of a simulation depending on how the environment is designed.

### 3.2.3. Role Plays

In role playing, individuals play pre-determined characters and invent the scenario in real-time, whereas simulation technology represents systems in which participants have functions [98]. Role plays are an efficient training method when trainers want to focus on specific training content and evaluate learner performance in a particular skill [76,99]. One potential explanation for the efficacy of role plays is attributed to the social and observational learning that can occur as a participant or an active spectator. Role plays accelerate how quickly learners integrate new knowledge and skills into behaviors [100]. Thus, a primary similarity between role plays and simulation technology used in workplace training is the scenario-based learning environment which challenges learners to apply their knowledge. A roleplay can be set within the context of a simulation depending on how the experience is designed.

### 3.3. Benefits of Simulation Technology

With respect to learning outcomes, we adopted the view of researchers who have asserted that instructional principles are more impactful than instructional mediums (i.e., computer, paper, video, audio) [16,101]. This is an important point in the discussion of potential benefits of simulation technology for training because the media, like the design, should be a function of the learning objective [102]. To illustrate this relationship, we explored how simulation technology facilitates various instructional principles.

### 3.3.1. Realism

The realistic nature of simulation technology is a salient feature and beneficial to learning outcomes [74]. Depending on the content and purpose of the training, simulation technology can provide physical fidelity (i.e., similarity in look, sound, and feel) and psychological fidelity (i.e., similarity in mental and/or emotional state) to facilitate learn-

ing [103]. The potential for high fidelity in simulation technology is critical to its efficacy as a learning medium because it bolsters retention via identical elements (i.e., the similarity between the training and the real-world application; [16,104]. Research on training transfer has suggested that skills are more likely to be reproduced in work environments that more closely resemble the training environment [67,105,106]. Simulation technology for workplace training enables instructors to create practice scenarios for nuanced behavioral skills, such as the stress management and adaptability required for empathic communication [76,107]. Given the traditional use of in-person instruction and the common use of computer-based training [53], simulation technology may help to bridge this gap by developing users' skills for lifelong learning outside of a formal learning environment. For learners using simulation technology, this can be considered an additional element of fidelity beyond focal knowledge or skill.

### 3.3.2. Practice Opportunities

Simulation technology for training can afford learners with opportunities to exercise control over their practice [77]. Control in simulation technology has been characterized as a learner's decision latitude over their strategies, role, and assumed responsibilities [6]. A constructivist view of learning suggests that using different vantage points to problem solve, create meaning, and apply knowledge may contribute to the creation of mental models [66]. High control simulation technology affords learners the latitude to isolate and repeatedly practice a skill, which can be considered effective practice [16]. Further, learner control over difficulty has been shown to have a strong moderating effect between self-efficacy and learning transfer in training which uses simulation technology [5]. In digital training settings, learner control and adaptive guidance have been shown to be effective design elements for complex skills by allowing learners to build on previous or fundamental knowledge [107]. Indeed, repeatedly practicing decision making in various roles and learning from mistakes have been reported as critical deep learning elements by learners using simulation technology [42].

### 3.3.3. Immersive Environments

Practicing using simulation technology is advantageous when a real-life environment is high stakes [5], meaning the tasks could be highly complex, have small margins of error, or present potentially dangerous or intimidating experiences [13,108]. The immersive elements of simulation technology used for training have been positively linked to effective learning procedures [108], which promotes simulation technology as an alternative to training scenarios where a mistake could result in serious injury, death, damaged equipment or other losses. For example, intimidating environments can add a degree of psychological stress to a work task [103]. Construction workers on a high-rise worksite need to demonstrate familiarity and comfort with safety procedures and tasks at a great height before being expected to do so in real life. Ultimately, utilizing simulation technology for training offers affordance to engage in error management training without the consequences of actual physical, emotional, or economic harm [82,109].

### 3.3.4. Feedback Capabilities

Simulation technology might be able to provide performance improvement feedback via behavioral metrics [6,14]. Detailed feedback can be integrated into simulation technology using techniques such as debriefing and metacognitive prompts, which have historically been linked to training transfer [27,43,72,110]. Developments in the field of computational psychometrics suggest that trace data (i.e., mouse movements/clicks, time spent on a task, selection choices) can be mined to predict individuals' cognitive ability or personality traits [111]. This preliminary evidence highlights the variety of data available to practitioners for use in providing mid- and post-training feedback. Additionally, learner metacognition has been highlighted as a fundamental state in the active learning pro-

cess [43]. In a review of simulation technology [6], the authors suggests that metacognitive interventions may be critical for knowledge outcomes in particular.

### 3.3.5. Cost as an Investment

Utilizing simulation technology for training may also provide long-term cost savings, despite the high upfront cost typically associated with well-designed solutions [54,79]. The flexible yet controlled environment of simulation technology allows organizations to develop standardized and customizable training programs that can mitigate the potential downsides of on-the-job training, such as untrained handling of equipment, premature execution of tasks by untrained or unskilled personnel, and potential worksite dangers and hazards which could lead to worker's compensation claims [2,112]. Thus, when evaluating the upfront cost of simulation technology as a training solution, it can be valuable to consider the opportunity for subsequent long-term cost efficiencies. In addition to the reduced maintenance costs, [113] found that one particular intervention, using simulation technology, produced cost savings by reducing administrative costs such as longer hospital stays. The costs of developing simulation technology for training can also be offset by overlaying innovative solutions such as augmented reality with existing training tools (e.g., medical manikins) or lower cost interfaces (e.g., tablets) [58,83]. Additionally, the skills required to develop simulation technology (i.e., coding, engineering, etc.) are becoming more available as STEM fields continue to grow [114]. The increasing availability of skilled programmers and product designers works to make simulation technology more accessible and affordable to organizations.

### 3.4. Challenges with Simulation Technology

Here, we highlight pertinent challenges to be aware of when considering using simulation technology for training.

### 3.4.1. Upfront Costs

There is a relatively high upfront cost associated with some simulation technology, usually associated with the time, expertise, testing, and revising involved in the development process. These costs can pose a financial risk if organizations do not realize the expected participation. Preliminary findings have suggested that a viable alternative to developing custom game-based interventions is to use commercially available off-the-shelf games, which are typically less resource intensive [54]. Since game-based learning shares attributes with simulation technology, it follows that commercial options may emerge as a path for simulations.

### 3.4.2. Variability in Learner Experiences

Secondly, there is variability in the applied knowledge, skills, and abilities for similar jobs and tasks. The breadth and ambiguity of learner tasks may create challenges for data collection in simulation technology [115]. This variability could result in a more resource-intensive process for instructional designers, workplace trainers, and educators since the efficacy of simulation technology as a training solution relies heavily on recreating real-life scenarios. To avoid this pitfall, organizations can consider using simulation technology for training generalizable skills versus specialized skills. For example, training on a forklift can be generalized based on the most common conditions (e.g., warehouse, side-loading, rough terrain, etc.), regardless of the specific forklift manufacturer. However, generalizable skills may still vary between individuals and the outcome of using simulation technology could produce results that would be difficult to mimic in everyday life.

## 4. Discussion

### 4.1. Modern Research Findings

Here, we provide insights on emerging research to reduce the challenges of rapid advancements. While this is not an exhaustive review, we did attempt to provide an

overview of recent research literature. The three themes we highlighted include design attributes, pedagogical challenges, and cognitive load effects (Table 4). These themes, drawn from the research review described above, extend the historical themes from Table 1.

**Table 4.** Linking Modern Themes in Simulation Technology with Historical Themes in Workplace Training.

| | |
|---|---|
| Design Attributes | Science of Learning: Establishment of parsimony has been an effective accelerator for similar training method research (e.g., serious games, game-based learning). |
| | Democratization of Knowledge: Availability of interdisciplinary research may lead to novel design attributes. Methodological studies are also critical to isolate and manipulate individual attributes. |
| | Scaling Productivity: Future research should explore design attributes which facilitate incremental observational and social learning compared to traditional methods. |
| | Individualized Learning: A taxonomy of simulation design attributes would enable educators to design interventions that take into account learner characteristics. |
| Cognitive Load | Emergence of Knowledge Work: Future research should continue to study the effects of cognitive load on simulation training transfer to improve its design efficacy. |
| | Democratization of Knowledge: Examining cognitive load effects on learning via simulation for training can mitigate potential adverse impacts to neurodiverse learner populations. |
| | Scaling Productivity: Simulations may simplify learning by reducing specific types of cognitive load compared to natural environments. |
| | Individualized Learning: All feedback is not created equal, especially in simulations with high fidelity. Research focused on the cognitive load effects of various feedback mechanisms will ensure simulations are designed effectively. |
| Pedagogical Challenges | Science of Learning: Industrial training effectiveness research may offer frameworks for building digital competencies. Additionally, higher learning institutions may realize additional benefits, such as educator job satisfaction and retention. |
| | Democratization of Knowledge: Pedagogical research will enable increased access to simulation-based learning and educator capabilities. |
| | Scaling Productivity/Individualized Learning: Simulation for training is not a substitute for traditional methods in most cases. Educators play a key role in adding feedback and meaning to learners' simulated experiences. |

### 4.1.1. Design Attributes

Researchers continue to reinforce the role of intentional design, methodology, and media selection to effectively facilitate learning objectives (e.g., [54,72,102]). Recent research has built upon existing instructional systems' design taxonomies and established theoretical implications for simulation technology design attributes in the context of training (e.g., [13,116,117]). For example, previous research has explored differences between gesture-based motions (i.e., using the thumb and index finger to zoom in/out of a screen) and mouse-based motions (i.e., clicking an icon which represents zoom) [118]. The authors found that young learners in the multi-touch, gesture-based group spent more time interacting with the learning content, likely due to the intuitive nature of the gesture-based motions. This is just one example of how modern advancements create room for next-generation design attributes in simulation technology. Delineating training design elements for use specifically with simulation technology parallels efforts to establish parsimony within the game-based learning literature. We expect to see more research on instructional design for simulation technology and encourage practitioners to familiarize themselves with emerging taxonomies.

### 4.1.2. Cognitive Load

Recent research in the medical field suggests growing interest in the linkages between cognitive load and learning outcomes in simulation technology used for training (e.g., [27,32,119–121]). The interaction of cognitive load in a learning environment using simulation technology is applicable beyond medicine. Previous research has found that

in-game metrics predicted cognitive load (and subsequently performance) within the first tenth of the game time [122]. This preliminary research is promising for the development of adaptive features associated with simulation technology in service of maximizing challenges while mitigating negative effects of overload. Similarly, other researchers have studied the effects of text annotation on cognitive load in the use of virtualized simulation technology [120]. In contrast to the cognitive theory of multimedia learning [123], the text annotations did not significantly reduce extraneous cognitive load. This finding was explained by the simple visual display of the learning material, such that learners did not need to divert cognitive resources to learning what had been provided in the display. We expect researchers and developers to continue narrowing in on how to leverage the benefits of simulation technology, including a potential for more deep learning and less cognitive load unrelated to the learning.

### 4.1.3. Pedagogical Challenges

Beyond the medical field [7,15], there is a lack of empirical guidance for integrating simulation technology into workplace training, which could slow work-readiness for those entering a workforce. However, there have been calls for increased digital competencies to enable organizations to meet continuously evolving demands. For example, the mass adoption of virtual training solutions challenged organizations to learn and execute technology-based training solutions. Industrially, there has been a paradigm shift from using digital tools for efficiency to using digital tools as an everyday necessity. International governments and organizations have begun to highlight specific digital competencies such as media literacy and digital content creation skills (i.e., programming), even addressing these as a key to economic growth [7]. In one previous research study, the authors present a framework which applies instructional design components to the requisite knowledge, skills, and abilities associated with using simulation technology in healthcare (i.e., simulation design, scenario design, simulation research, simulation program administration) [119]. Pedagogically, there appears to be consistent support that simulations are likely to be effective when used as a supplement to traditional instructional techniques (e.g., [6,120]).

### 4.2. Recommended Use of Simulation Technology

Based on the present review, we offer a direction on learning situations in which we recommend simulation technology. Here, we draw from previous researchers who mapped simulation technology learning outcomes to address specific behavioral, affective, and cognitive outcomes [6]. We add to this work by offering supportive empirical and theoretical research from the field of workplace training. Simulation technology has been shown to have a positive relationship with behavioral outcomes such as perceptual motor skills, teamwork, and social skills [6]. We strongly recommend using simulation technology for behavioral learning outcomes due to the transfer benefits associated with high fidelity learning environments [16,104,105]. Research suggests that training fidelity is critical when a skill is highly complex (e.g., self-regulation of cardiovascular response) [109] or when the physical environment is highly sensitive (e.g., operating a military aircraft) [4]. Additionally, simulation technology may be especially well-suited for affective learning outcomes such as motivation, self-efficacy, and training satisfaction [6,82]. We suggest using simulation technology for affective outcomes when practitioners have access to simulation technology solutions which evoke immersive experiences and afford learner control over the difficulty of the task. This recommendation may have a heightened impact based on meta-analytic evidence suggesting that learner control enhances the relationship between self-efficacy and transfer [5]. Finally, simulation technology has been shown to be particularly effective for advanced cognitive skills such as critical thinking, decision-making, and meta-cognitive learning strategies [53]. Thus, we recommend that educators consider utilizing simulation technology for training when a high-fidelity environment is requisite to the focal knowledge (e.g., physics, engineering). For example, simulation technology has consistently been shown to effectively develop spatial reasoning (e.g., [118,124,125]). Similarly, simulation

technology may be effective for procedural knowledge, rather than declarative knowledge, as a result of mental models' construction [120,126]. Beyond these use cases, simulation technology may not be a cost-effective training methodology for other cognitive outcomes.

*4.3. Limitations*

While this paper provided informed predictions about the future of simulation technology for training, we acknowledge that cutting-edge science in this field is rapidly advancing. This review offers a snapshot of the intersection of simulation technology and workplace training, but readers are advised to seek out the most recent research and use cases. We hope future empirical research on the application of simulation technology will continue to acknowledge relevant advancements in the field as a way to continuously bridge the gap between reviews. For practitioners, we also recognize that the pace of innovation and implementation may often outpace the pace of science. We hope the historical review provided in this paper provides context for evaluating the new features of simulation technology as they become available.

## 5. Conclusions

Simulation technology has emerged and evolved in a way that aligns with the advancing needs of training interventions in the workplace. From manual mechanisms to synthetic learning environments, simulation technology likely enables learners to interact with complex issues and apply their skills to real-world scenarios related to their discipline. While simulation technology used in workplace training shares similarities with serious games, game-based learning, and role plays in terms of using scenario-based learning environments, it differs in the degree of realism embedded in the design and delivery. Understanding these similarities and differences may help organizations develop a well-rounded training strategy which uses multiple modalities to achieve varying learning outcomes.

We also discussed the importance and application of evidence-based instructional principles when designing training using simulation technology. Integrating effective workplace training practices is critical for organizations seeking to reap the benefits of simulation technology, which include the improved transfer effects of immersive environments, high levels of learner control, and in situ feedback capabilities. Designing simulation technology for workplace training with instructional principles and a clear strategy may also provide long-term cost savings despite the high upfront cost of development. These points offer important insights into the potential knowledge, skills, and capabilities needed by professionals in learning and development, instructional design, or any other education-related role

Finally, emerging research was presented in an effort to equip practitioners with informed predictions to help reduce the challenging effects of rapid advancements in simulation technology. We highlighted three themes we expect to see more research on in the near future: design attributes of simulation technology for workplace training, cognitive load effects of using simulation technology, and increasing urgency for pedagogical support in higher education. Finally, we recommended using simulation technology for training primarily when a high-fidelity practice environment poses potential imminent risk to the learner, as well as for complex cognitive tasks and knowledge development.

**Author Contributions:** Conceptualization, D.R.S.; methodology, D.R.S., S.V.L. and A.R.; software, D.R.S., A.R. and K.K.; validation, D.R.S., A.R. and K.K.; formal analysis, D.R.S., A.R. and K.K.; investigation, D.R.S., A.R., K.K. and D.D.; resources, D.R.S., A.R., S.V.L., K.K. and D.D.; data curation, D.R.S., A.R. and K.K.; writing—original draft preparation, D.R.S., A.R. and S.V.L.; writing—review and editing, D.R.S., A.R., K.K. and D.D.; visualization, D.R.S., A.R. and K.K.; supervision, D.R.S.; project administration, D.R.S. All authors have read and agreed to the published version of the manuscript.

**Funding:** This research received no external funding.

**Institutional Review Board Statement:** Not applicable.

**Informed Consent Statement:** Not applicable.

**Data Availability Statement:** All data generated or analyzed during this study are included in this published article and its supplementary information files.

**Conflicts of Interest:** The authors declare no conflict of interest.

**Abbreviations**

The following abbreviations are used in this manuscript:

AI      Artificial Intelligence
ICT     Information and Communication Technology
VR     Virtual Reality
STEM   Science, Technology, Engineering, Mathematics

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
