# Peer review of "Reviewing Simulation Technology: Implications for Workplace Training"

_mti, doi:10.3390/mti7050050_

Round 1

Reviewer 1 Report

The "research review" on training simulation technology is relatively easy to follow and understand. However, taking into consideration the fact that we speaking of a journal paper here, I need to point out the following shortcomings: 

1. Vocabulary:

Throughout the paper: decide if you are speaking of "training simulation technology" or of "simulation technology" or of "simulation training".

Based on Google Scholar, it seems that "simulation technology" is a specific approach, inherent to higher education health fields. In addition, it seems that authors mostly speak of "simulation technology for ... training" but not of "training simulation technology". So, this is a crucial question to solve before any other changes as the vocabulary influences the way your readers understand your text.

In addition, when you speak of "simulation technology" and then "most modern simulations utilize technology..." then the exact meaning of the word "technology" in the context of your paper has to be defined - is it something that simulations use or is it something that uses simulations. And if you decide to use both these meanings then you need to carefully distinguish these two meanings, allowing your reader to easily grasp which meaning was used.

2. Structure of the paper

The paper gives an overview of the research area, but the paper does not follow the structure of an academic paper. Namely the paper lacks all the elements that could make the study repeatable. If you claim that you are reviewing the research on "..." then you should describe the methods you have used (how did you find your data, what were your search and exclusion criteria, who analyzed the data, what methods were used for analysis, etc.). 

From my point of view, the paper needs to be restructured and you would need to add the information about how you conducted your study (see above and below). 

=== Smaller questions === 

Abstract

The first sentence "Higher education has maintained a commitment..." should be rephrased as it claims something that could or could not be true in different contexts. For example, you could instead state that simulation technology has a special meaning for higher education because...

---

Introduction

Page 2, please review the sentence "The ongoing nature of rapid development provides a struggle to maintain 55 currency" for greater clarity (e.g., development of simulation technology - or simulated technology(?)).

---

Part I: Historical Themes...

In abstract you state that "we outline historical themes in workplace training in general that have influenced the development of simulation technology"

However, I argue that what you have done on pages 2-7 is excessive and more suitable for a textbook not a journal paper - as this is common knowledge, and indeed, it is too general for the scope of your paper. If you want to retain it in order to give your reader better perspective then, at least, you should compress the contents and make it more concise (in the context of your paper). It is a journal paper - you do not need to demonstrate what you know - you need to use only the most relevant information and trust your readers' abilities to independently learn more about the side-topics if they choose to. 

---

Part II: Current Research...

What you have done here, I find spot-on and highly relevant.

However, in order to avoid your paper being a report or precis of randomly chosen academic works, you should describe the methods you used for analyzing the "current research on training simulation technology". I.e., you should describe how you built your data (what databases you looked for current research for), how you analyzed it (how did you reach your overview of instructional methods?), etc.

In addition, it seems that the subtitles of Part II (e.g. "Serious Games for Instruction", "Game-Based Learning") could be a result of some analysis - you are describing simulation technology categories - how did you find them, did you use someone else's categories or did you code the papers in "current research".

---

Challenges With Simulations, Modern Research Findings

The problem here is the same as with the previous section - you list a lot of information without describing how you reached this information. In other words, the text lacks the elements of using the scientific method - Question -> Research -> Hypothesis -> Experiment -> Analysis -> Conclusion. 

---

Conclusion

In reality, this should be Discussion/Conclusions where you sum up and generalize your findings - but where you also put your work into the bigger context of your research area (comparison with the findings in similar works of other authors), discuss the limitations of your study, etc. These elements are all missing right now. 

Reviewer 2 Report

Overall, this paper is interesting and well researched but it suffers from structural problems.

Part I occupies almost 40% of the paper but its extensive historical overview is not reflected in either the title or the Abstract. It is also unclear how the overview informs Part II and subsequent sections of the paper.

My impression is that Part I and Part II have been written by different members of the authorial team, with insufficient attention paid to linkage of ideas and integration. Part II has merit and potential value to the readership, but the paper as a whole is currently unacceptable. These shortcomings are structural and can be rectified.

Before any further review is undertaken, and in order to justify the inclusion of Part I, I recommend the following substantial revision: (i) rewording the title and Abstract; (ii) reducing the size of Part I, and (iii) showing more clearly how the historical analysis informs discussion in Part II. The section numbering should also be revised, as only the Introduction is currently numbered and Parts I & II introduce further anomaly. I recommend a consistent renumbering throughout.

The revised paper should then be resubmitted for review.

The quality of Academic English is good.

Round 2

Reviewer 1 Report

I am glad to see that the authors have paid considerable effort and attention in order to address my comments that aimed at improving the work and making a worthy contribution to the field. 

Reviewer 2 Report

Thank you for your efforts to improve this paper in respect of its structural imbalance and coherence.

The historical review of simulations has been slimmed down but, to follow the metaphor, I think it's still a little overweight. However, the section has been well researched and it makes for an interesting read, so I don't recommend any further crash dieting.

I note the changes made to the Title, Abstract, Introduction and Conclusion sections. These have helped link the old Parts I & II at the front of the paper, but I would have liked to have seen this reflected in the Discussion at the back. Also, I think this section could have been improved by more discussion with reference to the considerable literature on user interface design and user experience – not least because these matters align closely with the Aims and Scope of the journal.

Finally, I welcome the consistent renumbering of sections.

In all, these changes have substantially improved the academic quality and consistency of the paper and, crucially, have made it more accessible and useful for  the readership. I recommend acceptance.